# What works in engaging communities? Prioritising nutrition interventions in Burkina Faso, Ghana and South Africa

Daniella Watson[1,2,3]*, Mary Barker[4,5,6,7], P. Romuald Boua[8], Samuel Chatio[9], Adelaide Compaoré[8], Marion Danis[10], Maxwell Dalaba[9,11], Agnes Erzse[12], Polly Hardy-Johnson[1,4], Sarah H. Kehoe[1,4], Karen J. Hofman[12], Wendy T. Lawrence[4,13], Engelbert A. Nonterah[9,14], Hermann Sorgho[8], Teurai Rwafa-Ponela[12], Kate A. ward[1,4,7], Aviva Tugendhaft[12], on behalf of the INPreP study group¶

1 School of Human Development and Health, Faculty of Medicine, University of Southampton, Southampton, United Kingdom, 2 SAMRC Developmental Pathways for Health Research Unit, School of Clinical Medicine, University of the Witwatersrand, Johannesburg, South Africa, 3 Department of Global Health and Social Medicine, King's College London, London, United Kingdom, 4 MRC Lifecourse Epidemiology Centre, University of Southampton, Southampton, United Kingdom, 5 School of Health Sciences, Faculty of Environment and Life Sciences, University of Southampton, Southampton, United Kingdom, 6 NIHR Southampton Biomedical Research Centre, University of Southampton and University Hospital Southampton NHS Foundation Trust, Southampton, United Kingdom, 7 School of Public Health, Faculty of Health Sciences, University of the Witwatersrand, Johannesburg, South Africa, 8 Clinical Research Unit of Nanoro, Institut de Recherche en Sciences de la Santé, Nanoro, Burkina Faso, 9 Navrongo Health Research Centre, Ghana Health Service, Accra, Ghana, 10 Department of Bioethics, National Institutes of Health, Bethesda, MD, United States of America, 11 Institute of Heath Research, University of Health and Allied Sciences, Ho, Ghana, 12 SAMRC/Wits Centre for Health Economics and Decision Science—PRICELESS SA, School of Public Health, Faculty of Health Sciences, University of Witwatersrand, Johannesburg, South Africa, 13 Primary Care, Population Science and Medical Education, Faculty of Medicine, University of Southampton, Southampton, United Kingdom, 14 Julius Global Health, Julius Center for Health Sciences and Primary Care, University Medical Center Utrecht, Utrecht University, Utrecht, The Netherlands

¶ Full list of INPreP study group members can be found in the acknowledgements.
* daniella.watson@kcl.ac.uk

**Data Availability Statement:** Data can be found on this link: https://doi.org/10.5258/SOTON/D2362.

**Funding:** This research was funded by the National Institute for Health Research (NIHR) (17\63\154)

## Abstract

### Background

"Choosing All Together" (CHAT), is a community engagement tool designed to give the public a voice in how best to allocate limited resources to improve population health. This process evaluation explored the mechanisms through which CHAT generates community engagement.

### Method

The CHAT tool was adapted and implemented for use in two rural communities (Nanoro, Burkina Faso, and Navrongo, Ghana) and one urban township (Soweto, South Africa) to prioritize maternal and child nutrition interventions. Community discussions were audio-recorded, transcribed, and translated into English. Twenty-two transcripts, including six each from Navrongo and Soweto and 10 from Nanoro, were analysed thematically to generate data driven codes and themes to explain mechanisms underlying the CHAT process. The process evaluation was based on the UK MRC process evaluation guidance.

using UK aid from the UK Government to support global health research. The views expressed in this publication are those of the authors and not necessarily those of the NIHR or the UK Department of Health and Social Care. KJH and AE are supported by South African Medical Research Council/ Centre for Health Economics and Decision Science – PRICELESS SA, University of Witwatersrand School of Public Health, Faculty of Health Sciences, Johannesburg South Africa (D1305910-03). The funders had no role in study design, data collection and analysis, decision to publish, or preparation of the manuscript.

**Competing interests:** The authors have declared that no competing interests exist.

## Results

Seven themes describing the functions and outcomes of CHAT were identified. Themes described participants deliberating trade-offs, working together, agreeing on priorities, having a shared vision, and increasing their knowledge, also the skills of the facilitator, and a process of power sharing between participants and researchers. Participants came to an agreement of priorities when they had a shared vision. Trained facilitators are important to facilitate meaningful discussion between participants and those with lower levels of literacy to participate fully.

## Conclusion

CHAT has been shown to be adaptable and useful in prioritising maternal and child nutrition interventions in communities in Burkina Faso, Ghana, and South Africa. Conducting CHAT in communities over a longer period and involving policy-makers would increase trust, mutual respect and develop partnerships.

## Introduction

The 2021 Lancet series on maternal and child nutrition recommends community engagement and mobilisation as a cross cutting strategy to improve maternal and child nutrition [1]. Community engagement and mobilisation have been key to addressing health issues, since Paulo Freire conducted his seminal work on the process [2, 3]. Through dialogue, those who are marginalised develop critical understanding of the social determinants of their health, their rights and a sense of collective agency to challenge social and health inequalities [4]. One method of engaging and mobilising communities is through involvement in deliberative priority setting. This includes involving representatives of communities affected by decisions and policies in their development. This process is based on dialogue that captures opinions, considers different points of views, promotes the possibility of consensus with limited coercion, deception or manipulation, and tends not to be tokenistic [5].

Choosing All Together (CHAT) is a deliberative community engagement activity designed to involve the public in decision-making where resources are constrained [6]. CHAT is designed as a board-game where players are allocated a pre-defined finite budget in the form of stickers that represent money. Then, through facilitated group discussion, they are encouraged to come to an agreement on the health packages for their community [6]. CHAT has been implemented in different settings and countries, including in low resource settings and with those who are disenfranchised from decision-making [7–9]. In one application of CHAT, minority groups from native and Arab American backgrounds prioritised health insurance packages differently based on having different cultural beliefs, which highlighted the importance of considering context and different opinions when seeking public input on spending priorities [10]. CHAT was also implemented with rural communities with low levels of literacy in Rajasthan, India, where the group came to an agreement on the priorities to be covered by health insurance for their community [11].

These studies found that participants enjoyed taking part in CHAT and valued being part of a decision-making activity [10, 11], yet there are no formal evaluation of the CHAT process that go beyond this. Butterfoss (2006) highlighted that there is poor understanding in general of how community engagement affects participants, or achieves its outcomes [12]. Evaluation

of the process of implementing community engagement tools can be used to develop more effective community-based interventions, encourage practical recommendations to implementation and to focus on how outcomes are achieved, rather than what outcome are achieved [12]. It has been proposed that the process of implementing community engagement interventions such as CHAT may be evaluated through answering a series of questions: who, how, when, why, how many, and how much community members participate [12]. Due to the range of actors, activities and processes involved in the development and implementation of community engagement interventions, such process evaluation is, however, known to be challenging [12].

Manafo et al (2018) describe deliberative priority setting as a two-way dialogue between stakeholders who discuss trade-offs and produce a set of agreed priorities derived through consensus [20]. Abelson et al (2003) developed a framework for evaluating deliberative priority setting which includes representation of participants, achievement of consensus or agreement, legitimacy and accountability and level of engagement [13]. Drawing on the wider community engagement literature, Butterfoss (2006) emphasises the importance of also assessing the role of the researcher, satisfaction with the process of participation and balance of power and leadership [12].

At the time of the study described in this paper, CHAT had not been implemented with community members in Burkina Faso, Ghana, or urban South Africa, nor been used to prioritise nutrition interventions; the way CHAT works in these contexts and with this topic was not understood. Our research team, the Improved Nutrition Preconception, Pregnancy and Post-delivery (INPreP) [14], were the first to adapt CHAT for those specific populations and for nutrition interventions. INPreP aimed to identify feasible and appropriate maternal and infant nutrition interventions through community engagement in Burkina Faso, Ghana and South Africa [14], implementing CHAT to understand the community members' programme priorities for improved nutrition. In Burkina Faso, the community members prioritised women's capacity through agriculture, mass communication about nutrition, and child-friendly school initiatives [15]. In Ghana, the community members prioritised livelihood empowerment, male involvement and micronutrient supplementation [16]. In South Africa, the community members prioritised school breakfasts, six-months paid maternity leave, and improved food safety in South Africa [17]. This paper adds to the community priorities from the above publications and addresses the gap in evaluation of community engagement approaches such as CHAT, which Butterfoss (2006) highlighted as important to understand how community engagement works [12].

This paper addresses the following research question: How well does the CHAT process work in engaging and prioritising nutrition interventions for communities in Burkina Faso, Ghana and South Africa?

## Materials and methods

We used the United Kingdom (UK) Medical Research Council (MRC) guidance on process evaluation of complex interventions to identify the mechanisms of impact of CHAT and to consider how context and implementation influenced the CHAT process [18]. It is important to understand the three different contexts and how CHAT was implemented by the three different research teams, which can ultimately affect the mechanisms of impact of CHAT.

We describe the context of the three settings in the methods section to give an overall background of the economic, health and social factors that might influence the mechanisms of impact of CHAT, based on published data and the research teams experiences of working and living in these settings. We also describe the implementation of CHAT in all three settings to

highlight the similarities and differences in design and implementation of CHAT by the research teams. The description of CHAT implementation is based on the INPreP research team meeting updates about the progress of CHAT and the challenges. The mechanisms of impact of CHAT were identified through the qualitative analysis of the transcripts of the CHAT activity, described in the results section of the paper. Although CHAT is not yet considered to be a complex intervention, using the MRC process evaluation guidance to evaluate community engagement is novel and is appropriate to evaluate the processes involved.

Data on the CHAT process were collected in Nanoro in Burkina Faso, Navrongo in Ghana in December 2020 -February 2021, and Soweto in South Africa in September 2021. Data was captured through audio-recordings of participants' discussions whilst participating in CHAT. Recordings were intended to capture the participants' thoughts, motivations, and challenges in 'real-time'. The audio-recordings were transcribed and translated from local languages into English: More into French then English in Nanoro; Kasem and Nankam into English in Navrongo and; Xhosa, Zulu into English in Soweto. Qualitative analyses of the transcripts from the three settings were analysed by the research team to explore how CHAT was conducted in each setting. Analysis of transcripts from the CHAT pilot studies are not included in this paper. Data were analysed thematically using NVivo 12 software QSR International to organise the text [19]. Transcripts were analysed inductively to ensure the findings were data-driven and with a level of deductive analysis based on previous research on deliberative priority setting and community engagement [12, 13, 20]. A coding framework (S5 Appendix) of the themes was sent to researchers at each site who gave suggestions on how to frame the analysis; the coding was discussed at regular team meetings in order to include perspectives from those who conducted CHAT in each setting. The consolidated criteria for reporting qualitative research (COREQ) guidance were used to systematically report the qualitative methods [21]. The authors did not have access to information that could identify individual participants during or after data collection.

This study was conducted according to the guidelines laid down in the Declaration of Helsinki and all procedures involving research study participants were approved by Faculty of Medicine Ethics Committee, University of Southampton, UK (47290), the University of the Witwatersrand Human Research Ethics Committee (Medical), South Africa (M181056), the Navrongo Health Research Centre Institutional Review Board, Ghana (NHRCIRB322), the National Health Ethics Committee in Burkina Faso, (201 8-12-156). Written and verbal informed consent was obtained from all participants. Verbal consent was witnessed and formally recorded.

## Context

**Nanoro, burkina faso.** CHAT discussions were carried out in low income, rural communities in Nanoro and Soaw; these districts comprise 24 villages located in the Centre-West region of Burkina Faso, 90 km from the capital city, Ouagadougou. In 2017 it was estimated that 171,000 people lived in this area and were mainly Christian or Muslim from the Mossi ethnic group [22]. Fifty five percent of people live on less than GBP 0.50 per day [23] with the main source of income being subsistence farming which is dependent on seasonal fluctuations in climate and rainfall. Rearing livestock is the second economic activity in this population, much of which is sold [23]. Households average eight people, mainly consisting of a husband, a wife, and their children [22]. Women head 10% of the households, usually because they have been widowed or abandoned by husbands who have migrated [22]. There is a high prevalence of stunting (25.5%), wasting (9.0%) and underweight (16.4%) in children under five years old with 2.6% of under-fives being overweight [24]. Communicable diseases such as malaria, acute

respiratory infections and fever of unknown origin are still the leading cause of mortality among adults and children [25, 26] with poor sanitation and inadequate potable water supply exacerbating these health issues. The study communities are served by seven peripheral health facilities and one district hospital (Centre Médical avec Antenne Chirurgicale) [27]. Some of the challenges faced by this community include agriculture issues, managing infectious diseases with limited health access, and socially and culturally determined gender roles restricting what women are permitted to do [28].

**Navrongo, Ghana.** Navrongo hosts the Navrongo Health and Demographic Surveillance Systems (HDSS) whose coverage area includes the Kasena-Nankana East Municipality and Kasena-Nankana West District of Northern Ghana. The HDSS area covers an area of 1675 km$^2$ of Sahelian savannah with a population of about 153,000, which is increasing [29]. There are two dominant ethnicities (Kasena and Nankani); communities are mainly Christian, with a Muslim minority. The main source of income is subsistence farming which is dependent on seasonal fluctuations in climate and rainfall. The population largely lives on subsistence crops including millet, sorghum, rice, maize and groundnuts, alongside some fruits and vegetables [30]. The main sources of water are streams, wells and boreholes [31]. Most people live in multi-household compounds: 54% live with 1–5 people, 24% with 6–10 people, 17% with 11–15 people and 5% with 16–20 people [32]. There is a high prevalence of stunting (14.2%), wasting (6.8%) and underweight (12.6%) in children under five years old with 2.9% of under-fives being overweight [33]. The health system comprises primary, secondary and tertiary health care delivery with two secondary districts hospitals located in Navrongo and Paga and a tertiary regional hospital close by in Bogatanga, the regional capital [34]. The Community and Health Planning Service (CHPS) centres which form the primary health care system in Ghana has stationed community nurses as well as health volunteers, who are community members. This system has been instrumental in reducing infant mortality and fertility rates [35]. Infectious diseases, such as malaria, diarrheal diseases, acute respiratory infections, and immunisable diseases, are the predominant cause of deaths but are declining [36, 37]. Some of the challenges facing this community include poverty, lack of irrigated agricultural land and poor harvests [38].

**Soweto, South Africa.** Soweto is a rapidly transitioning urban township located on the outskirts of Johannesburg, with a population of approximately 1,695,000 [39]. Soweto is mostly populated by black South Africans. In a 2018 survey, 50% of Soweto respondents were unmarried and 40% unemployed [40]. Many ethnic groups reside in Soweto including Zulu, Sotho, Tsonga, Tswana, Xhosa [41]. Many children are raised in single parent families [41]. Exposure to violence for women and children is common [42]. Conditions in lower income neighbourhoods are poor, houses have tin roofs, shared public toilets, unpaved roads, and few trees [43]. Multiple families share a single fenced property, with additional smaller rooms or shacks built on [43]. There is a lack of access to sanitation in informal settlements due to the rapid rate of household formation and reduced space for onsite toilets [44]. The food environment in Soweto has been described as obesogenic as it is dominated by street vendors selling cheap high-sugar snacks and deep-fried processed foods, and advertising of high-energy, processed food and beverages, especially those with added sugar [45]. There is high prevalence of children under five being both overweight (12.9%) and stunted (23.2%), with underweight and wasting at 5.5% and 3.4% respectively [46]. Health challenges in Soweto include high rates of HIV/AIDS infection, tuberculosis, rising rates of diabetes, infant and maternal mortality, road traffic incidents, intimate partner crime, rape and murder [47]. Africa's largest tertiary hospital, Chris Hani Baragwanath Academic Hospital, is in Soweto. Members of these communities are preoccupied by issues of poverty, unemployment, the obesogenic food environment and challenges related to alcohol, substance and domestic abuse and violence [48].

## CHAT implementation

The CHAT tool and accompanying manuals were modified and implemented in rural Burkina Faso and Ghana and urban South Africa based on our team's research in rural South Africa [49, 50]. The process included identifying nutrition interventions during formative discussion groups[28, 38, 48, 51–53], costing these interventions, designing the CHAT boards (Figs 1–3), developing separate intervention manuals for the research teams and participants, training research teams, and piloting and then implementing CHAT. The CHAT boards have the numbers of dots in each intervention segment that represent the cost of the type of intervention relative to the total budget. The CHAT manuals for participants had a short description of the intervention packages with an image (S1–S3 Appendices). The CHAT manuals for the research teams were standardised for all three sites but adapted to each setting. The facilitator facilitated CHAT by reading the manuals like a script which included CHAT instructions and questions to ask the participants (S4 Appendix). CHAT was facilitated by experienced INPreP researchers ranging from those who were completing or had completed Masters or PhDs (AC, RB, MD, SC, TR, AE). Led by the facilitator, groups of participants played rounds of CHAT allocating stickers to represent the cost of the nutrition interventions that they wanted to prioritise. To reach agreement on the nutrition interventions to prioritise, participants had to deliberate and work together. The full description of this process is published elsewhere [16, 17].

Facilitators and researchers were trained on how to conduct CHAT by MD, AT, AE and in facilitation skills based on 'Healthy Conversations Skills' training by DW and PHJ [54]. CHAT was piloted in each setting to allow the facilitators to practise delivery, and to test the acceptability of the intervention and the process with the participants [55]. The teams made changes to the CHAT activity based on the participants' feedback on the pilot round. This included changing stickers in Nanoro and Navrongo, adapting the CHAT board icons in Nanoro, and giving more time to help explain study materials to participants in Soweto. Participants were encouraged to provide alternative and conflicting opinions as part of this process.

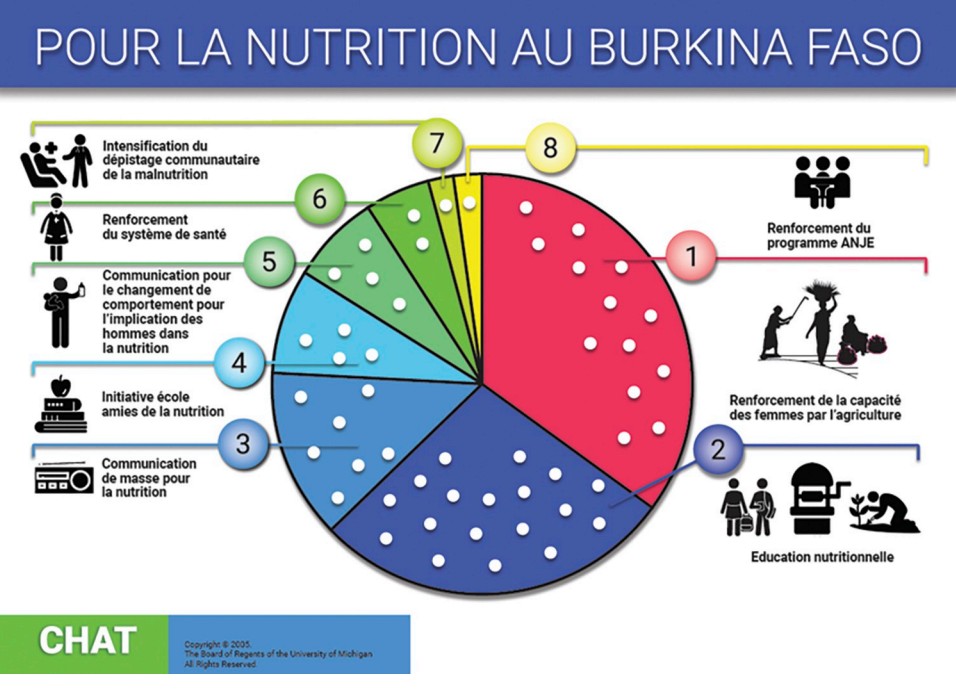

**Fig 1. Nanoro, Burkina Faso CHAT board.**

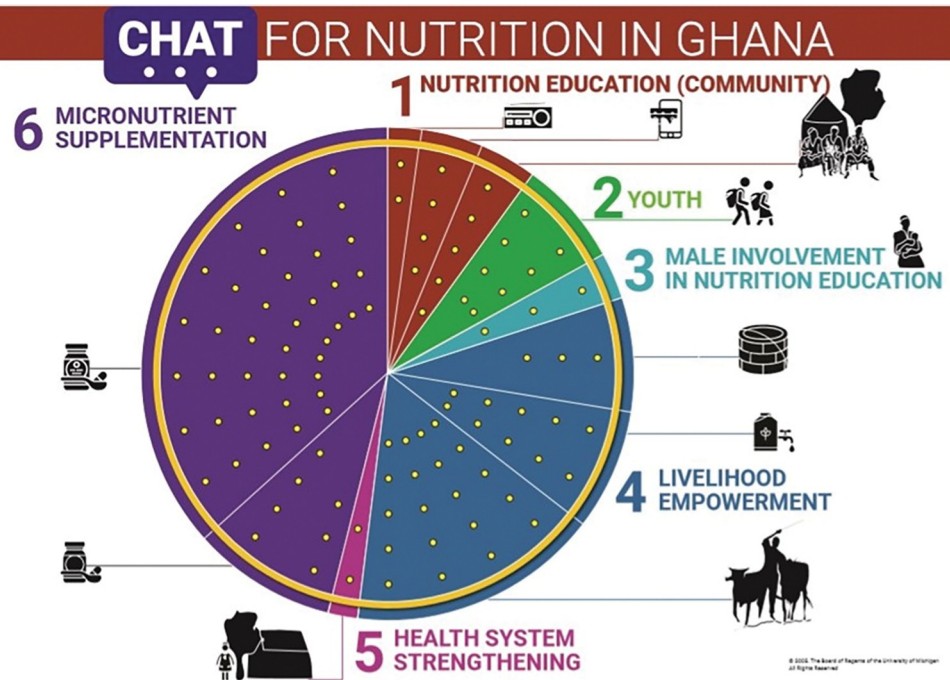

**Fig 2. Navrongo, Ghana, CHAT board [16].**

**Nanoro CHAT implementation.** Ten CHAT activities were conducted in ten villages (n = 93, aged 19–55 years). Two were conducted with men's groups (total n = 31) and eight with women's groups (total n = 62). Participants were selected with the support of community informants who explained to the potential participants the objective of the study and asked them, in person, if they would like to participate. If the potential participant was interested, the community informant organised an appointment for them to participate in CHAT. CHAT

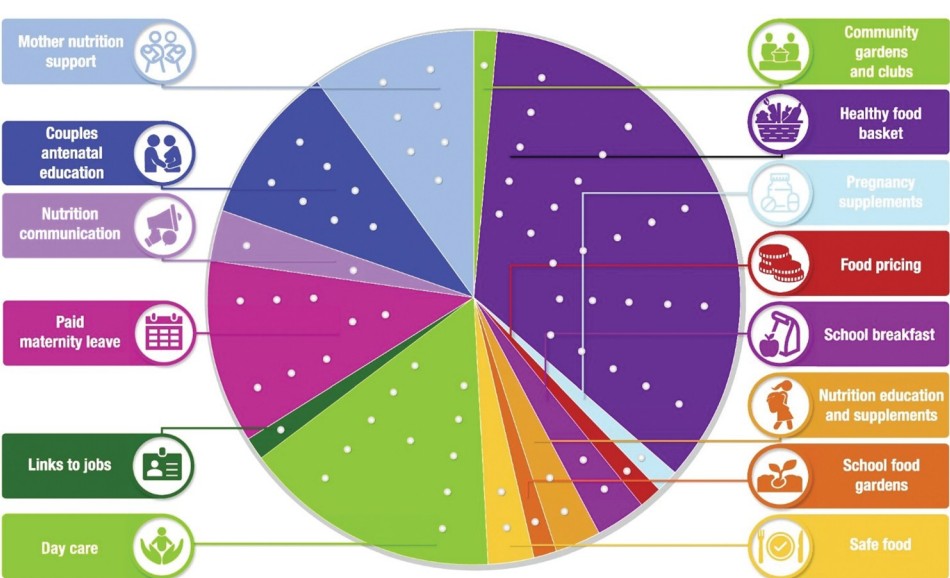

**Fig 3. Soweto, South Africa, CHAT board [17].**

was facilitated by one facilitator and supported by three researchers who took consent and helped participants with reading and understanding the CHAT materials. In total, each CHAT activity took a whole day to complete including time taken to transport participants to and from the venue. The venue was chosen by the research team to be a practical and comfortable space for the participants. Challenges of organising CHAT included clashing with market day and supporting the participants to understand what they were being asked to do. Participants appeared at first to be intimidated but with time and support seemed to find the CHAT process fun and engaging.

**Navrongo CHAT implementation.** CHAT was conducted in six communities in one group per community (n = 53, aged 24–50 years). The Navrongo HDSS database was used as the sampling frame for the selection of participants. Each CHAT activity had between 6–10 participants both men and women. One facilitator supported by three researchers helped participants to read manuals and gained consent from the participants. The venues, mostly in a central location within each community, were chosen with the aim of making the participants feel comfortable and to reduce travel time and costs. Challenges observed included participants being late to the activity, causing others to have to wait for lengthy periods. There was also a need to read and explain at length the intervention manuals. Having researchers on hand to explain the interventions in detail was useful. The CHAT activity took 2–3 hours, which left participants feeling tired. Extreme weather conditions disturbed some aspects of the activity; on some occasions strong winds blew away materials during the discussions.

**Soweto CHAT implementation.** CHAT was conducted with six groups (n = 47, aged 18–50+ years), including two men's groups (n = 16), two women's groups (n = 13), and two mixed men and women's groups (n = 18). Groups were conducted with men and women of similar age ranges within the same group. Research assistants recruited participants, who were waiting at the Chris Hani Baragwanath Hospital taxi rank, if they would like to take part in a study. If they were interested, they were invited to take part in CHAT. This method of recruitment has been used on previous occasions by the research team [48, 53]. The research team attempted not to include participants who knew each other within the same CHAT group in case this influenced the discussion. Participants mainly used public transport to get to the venue, a research centre based at the hospital in Soweto. The cost of travel was reimbursed. A team of four researchers and research assistants were present, one acting as facilitator, another as observer, and the others to support the participants with the CHAT manual. Issues recorded included participant responses that were difficult to hear because of the masks that had to be worn to comply with COVID-19 protocols. Windows also had to be kept open for air circulation to meet COVID-19 protocols. This meant that participants were cold and discussions were disrupted by background noises.

## Results

### Mechanisms of impact

A total of 22 transcripts were analysed: six transcripts each from Navrongo and Soweto and 10 from Nanoro. This consisted of 22 CHAT activities with 193 participants. Themes are presented in Table 1 and S5 Appendix. The themes reflect potential mechanisms of impact through which CHAT may achieve its outcomes; these are drawn from the deliberative priority setting and community engagement literature described in the Introduction of this paper. These themes are presented with illustrative quotes from each site by gender and age group.

**1. Deliberating trade-offs.** For the priority-setting component of CHAT, participants deliberated and weighed interventions against one another. The discussions demonstrated

**Table 1. Identified themes and ways that CHAT works.**

| Mechanisms of impact |
| --- |
| 1. Deliberating trade-offs |
| 2. Agreement on priorities |
| 3. Working together |
| 4. Having a shared vision |
| 5. Increasing knowledge |
| 6. Having a skilled facilitator |
| 7. Sharing power |

how they considered the cost of each intervention, the value of the intervention to their community and how much of the community fund was left.

> *It's because we didn't have enough money otherwise, we would have bought it all.* FGD07_Women_35-55yrs_Nanoro, Burkina Faso

> *What I understand is that the interventions we have seen are those we are going to buy with our money with the stickers given to us and those that we choose are those we need most in our families or community.* FGD05_Men_24-34yrs_Navrongo, Ghana

> *When we checked our prices and other things, we had to make changes so that it can achieve what we wanted.* FGD18_Men_18+_Soweto, South Africa

**2. Agreement on priorities.** Another key component of the CHAT process described was that the participants came to an agreement on which interventions to prioritise. To do this, the participants engaged in discussion and meaningful dialogue. At the end of the discussion, the participants largely expressed satisfaction with the group decision and the agreement reached.

> *We are satisfied with our group choice.* FGD10_Men_18–34_Nanoro, Burkina Faso

> *For my side I think there are no modifications because those are what we want, and we also selected them based on our needs in the community.* FGD05_Men_24-34yrs_Navrongo, Ghana

> *We had come to an agreement and this thing was resolved and we thought of how we could change it* [nutrition of mothers and children]. FGD18_Men_18+_Soweto, South Africa

**3. Working together.** The participants demonstrated and discussed the importance of working together to solve a problem when selecting nutrition interventions. They felt the only way to solve health issues was by working as a group. This included listening to each other's opinions and considering each other's different backgrounds.

> *If we work together and there is help, I know that there will be change.* FGD09_Women_35-55yrs_Nanoro, Burkina Faso

> *Working together will let us get what we need without any misunderstanding. I think as we all come together as one and make our choices, we will still continue to work hard to achieve our aim as a team.* FGD05_Men_24-34yrs_Navrongo, Ghana

*I think the group setting helped us a lot because we come from different places and backgrounds so we can elaborate things to you, and it also shows you the other side of the coin which you might not know so it helps a lot because we learn from each other.* FGD19_Men_50+_Soweto, South Africa

**4. Having a shared vision.** Participants spoke of common issues in their community and ways to solve them collectively to bring about change. Examples from the community included: tools to harvest in Burkina Faso; to look out for each other to collectively get nutritious foods to mothers and children in Ghana; and more employment opportunities in South Africa. The community members in each setting seemed to share the same vision on how to solve these issues.

*When we have the work tools, we will be able to harvest more than before. There will be more to eat, there will be joy and peace in the household. . . . As far as we are concerned, I don't think that anything could hinder the implementation. Our whole community needs this help. Nobody could say that they did not need this help.* FGD08_Women_18-25yrs_Nanoro, Burkina Faso

*What I think is that if everybody has his/her hand in work to do and nobody is looking up to another person for support each day and night, it will help us to bring change in our lives because that will help your wife, the children and yourself to get nutritious food to eat every day.* FGD04_Men_35-50yrs_Navrongo, Ghana

*It becomes easier if you see this kind of things in our everyday life and not every one of them is easy but some of those decisions were not that hard to make because it is what we see daily.* FGD21_Mixed_18+_Soweto, South Africa

The participants also started to make plans to support their community to work together following the CHAT process. This suggests that the process of coming together and discussing priorities might lead to lasting change in the way the communities worked to solve their problems.

*As far as we, the men, are concerned, we can put little groups in place, and we will choose people to be responsible for organising the work. Everyone will be able to have their field, however, we can identify 2 hectares of space for the youngsters; in this space we can have a variety of crops. We can sew corn, sesame, artichokes. We are going to let the community leaders accompany them.* FGD13_Women_26-34yrs_Nanoro, Burkina Faso

*When we leave here it does not mean everything ends here. We will still have to meet as a group and discuss how we will be able to eat nutritious foods and it will also serve as opportunity for people to share with us how they get their healthy foods but if we leave here and everything ends here then I do not think we can improve nutrition in this community.* FGD03_Women_26-39yrs_Navrongo, Ghana

*I think uhm. . . we must, together with the community, do the community garden, so that we can be able to help other people who are not working.* FGD20_Women_18–49_Soweto, South Africa

**5. Increasing knowledge.** Participants felt their knowledge about nutrition had increased as a result of being involved in the CHAT process. The participants wanted to share this new knowledge with community members who were not part of the CHAT activity.

*I'm an old lady so it was difficult. [Laughs] When we got the cards, I didn't understand anything. I just wanted to take the stickers and line them all up and say that, we paid, and that's it. [Laughs] But regarding the explanations, I could understand. If I go back home, I'm going to say to my daughters-in-law to come and I will teach them.* FGD09_Women_35-55yrs_Nanoro, Burkina Faso

*What I have to say is that, what you people taught us here, when we go home we should also teach our colleagues and we come together as one team to work, to improve nutrition for the community.* FGD05_Men_24-34yrs_Navrongo, Ghana

*It was difficult, but we learnt something, and we will be able to help others with problem solving.* FGD22_Mixed_18+_Soweto, South Africa

**6. Having skilled facilitators.** The facilitators were key to implementation of the CHAT process. Analysis of the transcripts showed that it was important for participants' understanding of the process that facilitators checked in with them.

Participant: So, when we choose an activity, we stick it all down and together we keep choosing one and then we stick it down?

*Facilitator: Yes, but you must all agree, because you could choose and your partner might not have the same opinion as you because for her, it isn't important. So, you have to convince each other. But you can choose and change your minds. You can unstick* [the sticker] *and stick it down on another activity.* FGD09_Women_35–55_Nanoro, Burkina Faso

*Facilitator: What is the aim of this activity?*

*Participant: The reason why we are here is to discuss what we can do to be able to take care of our families and what we can do to get something to eat. Those are the reasons why we are here to discuss and get solutions for healthy life in our jurisdiction. This will make us grow strong, healthy and have the strength to do whatever we want.* FGD04_Men_35-50yrs_Navrongo, Ghana

*Facilitator: So, you do know what food pricing means in your manual, right?*

*Participant: Yes ma [Mrs].* FGD17_Women_50+_Soweto, South Africa

The facilitators were trained in Healthy Conversation Skills[54]. They demonstrated that they used key skills such as asking Open Discovery Questions starting mainly with 'How' and 'What'.

*Facilitator: But how are you going to do that, to choose the project which will be the most beneficial for you?* FGD07_Women_35-55yrs_Nanoro, Burkina Faso

*Facilitator: What parts of the programme would work and what parts of the programme wouldn't work?* FGD05_Men_24-34yrs_Navrongo, Ghana

*Facilitator: What are the consequences of not having a job and not being able to put food on the table? How does that affect family units and the community at large?* FGD22_Mixed_18+_Soweto, South Africa

**7. Sharing power.** The power dynamics between participants and facilitators was clear from the way the facilitator led all activities and directed discussions. This was also evident in the way participants thanked the facilitators for sharing the CHAT activity with them and

spoke of how the research team might help them in the future. In this way, the community members perceived the power to be in the hands of the facilitator more so than their community. In Burkina Faso and Ghana, the participants emphasised how the research team enlightened them and increased their knowledge, which implies that the power was with the research team over the participants. In Soweto, they spoke more of the other stakeholders not in the room, who have power but are not listening to them.

> *Yes, it can help. The fact that you have taken us here and enlightened us, we can follow it.* FGD09_Women_35–55_Nanoro, Burkina Faso

> *We are like blind people and you people came to open our eyes. We think today will not be your last coming to this community. If you people continue coming to educate us, we will then learn how to improve nutrition in [NAME OF COMMUNITY] and start our own nutritional programmes for development of the community.* FGD05_Men_24-34yrs_Navrongo, Ghana.

> *I am also grateful to people that are like me. . .that like to speak because we like to speak so that we do not bottle things up. But, we cannot as there is no-one who asks us, what do you want and what is happening. Things that bother us are always taken as a "political" issue but they do not come to us and sit with us, unless there is an issue to be resolved—unlike, how we are sitting today and discussing what is happening around us. They do not speak to us.* FGD17_Women_50+_Soweto, South Africa

## Discussion

This cross-cultural evaluation explored how people from three communities in Burkina Faso, Ghana and South Africa participated in activities using the community engagement and deliberative priority setting tool, CHAT. This discussion on lessons learnt will consider how the context and implementation of CHAT intertwined with mechanisms of impact identified from the qualitative analysis [18]. Seven themes were identified that indicated the potential ways in which CHAT operates to support participants to reach an agreement on prioritisation of nutrition interventions. These were deliberating trade-offs, agreement on priorities, working together, having a shared vision, increasing knowledge, having a skilled facilitator, and sharing power between participant and facilitator.

These themes align with the literature on deliberative priority setting and community engagement described by Butterfoss (2006) [12] but less to Abelson et al (2003) [13]. We perceive that CHAT succeeded in facilitating deliberation and identifying nutrition interventions through engaging in trade-offs, agreements and two-way dialogue [5]. Also important considerations were observed in CHAT about the role of the facilitator and the imbalance of power between facilitator and participants, which Butterfoss (2006) also emphasises [12]. Less was observed and analysed on representation, legitimacy and accountability and level of engagement, which are features Abelson et al (2003) described to evaluate deliberative priority setting [13]. This might be because we didn't ask the participants about these levels of engagement, that these concepts are more difficult to observe or that CHAT has limitations in these areas.

Other factors not directly captured in the community engagement literature was that participants worked together and had a shared vision. This could be interpreted as evidence of the psychological concept of collective efficacy, which Bandura described as a person's "shared beliefs in their collective power to produce desired results" [56, 57] p75. This work suggests that community engagement approaches such as CHAT can potentially increase participants' sense of collective efficacy to work together on issues that they care about and spark ways of

solving the issues together. Collective efficacy could be considered an integral part to community engagement approaches including CHAT. Although collective efficacy was not directly measured in this study, Bandura does consider reaching a group consensus or agreement as one of the markers of collective efficacy [56, 57], which was demonstrated consistently in CHAT.

### 'Having a skilled facilitator'

'Having a skilled facilitator' was important for understanding the participants contexts and also the CHAT implementation. The research team reported that implementation of CHAT was successful in each setting and the participants said that they enjoyed taking part. The role played by the facilitator seemed to be important factor of CHAT. The facilitator engaged the group, managed the group dynamics, and steered the CHAT process in the direction of the research aims. Other community engagement approaches have also recognised the role of the facilitator and identified key characteristics such as having motivated, trusted, local women, who might be integrated in the health system such as community health workers [58]. They also require specialist training to facilitate effective meetings and ensure inclusion of the most vulnerable and to enable prioritisation [58]. Our team provided specialist training to facilitators through Healthy Conversation Skills training [54].

The research teams were able to implement CHAT in each setting with micro contextual interferences such as weather conditions in Ghana, availability of participants on certain days due to the market in Burkina Faso and COVID-19 protocols affecting the CHAT process in South Africa. This shows the importance of 'having a skilled facilitators', who understands and are familiar with their context well enough to adapt to the macro-and-micro contexts of their participants when implementing CHAT.

In the three settings, the level of literacy was a well-considered contextual factor by our research team, especially in Navrongo and Nanoro. Fishkin's work on deliberative democracy suggests that the quality of the information provided to participant is key to deliberative processes [59]. Early in the CHAT modification process, our research team who have worked with the communities over many years, identified that the information being presented needed to match the participant's literacy levels. Our team developed interactive context-specific materials with illustrative graphics, ensured there were multiple researchers available to help participants read the materials and spent at least half a day on CHAT to ensure there was enough time for questions to be answered to enhance understanding. We wanted those of lower levels of literacy to feel comfortable enough to speak freely and ultimately increase the power of the participants to agree on priorities. In low-resource settings, researchers should ideally work with communities or teams who know them and at least conduct formative work to ensure that the deliberative priority setting process is appropriate. Again this shows the importance of skilled facilitators and research teams who know the contexts of their participants.

### Implications

Although the qualitative analysis demonstrated that CHAT can be conducted in the three settings, consideration should be made as to whether CHAT was the most appropriate tool to use with the communities in Burkina Faso, Ghana, and South Africa. It could be argued that community engagement interventions should have a more responsive rather than didactic facilitation style to reduce unbalanced power dynamics. The analysis showed that there was power imbalance within the CHAT activity between the participants and facilitators, where the facilitators led the activity determining who could talk when and directed the discussion according to the pre-set research aims. Community engagement interventions should actively monitor

and evaluate the power dynamics between community members and the wider research team throughout the design and implementation. To be truly empowering, community members should have a more active role in setting the agenda for the discussions, developing the CHAT board and leading the discussions. This has been demonstrated in programmes with young people through youth-led committees and councils [60].

Engaging community members with CHAT on one occasion only, without any follow-up support, has ethical implications. Inviting community members to engage in these discussions may raise their expectation that change is coming, yet a single CHAT activity does not guarantee that. Engaging community members on a single occasion is considered tokenistic like a tick box exercise [61]. This might imply that we consult, extract the data, and then disappear, instead of working alongside the communities over time to develop relationships and trust to support their problem-solving. To make the CHAT engagement more useful, we could engage policymakers in CHAT or feed back the communities' priorities to the policymakers as they have the power to potentially implement the interventions. Other approaches to community engagement and mobilisation, such as Participatory Learning and Action groups, engage with communities monthly over several years [62]. This builds up a partnership between communities and research teams and increases the capacity of the communities to bring about their own change, which is more sustainable in the long-term and does not necessarily require additional funding.

## Strengths and limitations

The evaluation was constrained by COVID-19 pandemic restrictions. During this time, we had both travel restrictions and extra COVID-19 work to balance with conducting CHAT. Interviews with and in-person observations of CHAT participants and facilitators to explore their experiences might have given a more in-depth understanding of whether CHAT was acceptable and how it works. Yet our teams did not have the capacity to conduct these methods at the time. In Soweto participants were recruited using an established recruitment strategy which has been used in previous research and has been effective in capturing the diversity of sociodemographic characteristics of Soweto's inhabitants [48, 53]. The approach involved approaching prospective participants at a hospital taxi rank and it is possible that the entire community was not represented using this method. Despite these challenges, a substantial qualitative analysis of 22 CHAT transcripts was conducted. The wider INPreP team supported the analysis which elicited a wider interpretation and increased cultural relevance.

## Conclusion

It was the first time that CHAT was implemented in an urban township in South Africa, in two rural African settings in Ghana and Burkina Faso and adopted to prioritise nutrition interventions. These unique applications show the adaptability of CHAT with different populations and health topics. Although CHAT can be implemented with success, consideration should be given to some of the challenges including that of engaging community members on a one time basis as well as the power dynamics between participants and the research team. One potential solution is to work with community members and policymakers using CHAT longitudinally, to build rapport and to understand their changing priorities over time. This paper contributes to the wider community engagement literature by outlining key features demonstrated in CHAT. As well as raising considerations about power dynamics, this paper also compliments the existing literature about the importance of participants deliberating trade-offs, agreeing on priorities, and having a skilled facilitator. This paper also adds that when participants have a

shared vision, they are more likely to agree on priorities and even take these actions forward with their community after participating in CHAT.

## Supporting information

**S1 Appendix. CHAT participant manual Nanoro, Burkina Faso.**
(DOCX)

**S2 Appendix. CHAT participant manual Navrongo, Ghana.**
(DOCX)

**S3 Appendix. CHAT participant manual Soweto, South Africa.**
(DOCX)

**S4 Appendix. CHAT facilitator training manual.**
(DOCX)

**S5 Appendix. CHAT Evaluation coding framework.**
(DOCX)

## Acknowledgments

The authors would like to thank the participants for their time and interest in the study, and the field workers who conducted CHAT. We would also like to thank the INPreP group: Abraham Oduro, James Adoctor, Paul Welaga, Paula Beeri, Edith Dambayi, Esmond W. Nonterah, Winfred Ofosu, Doreen Ayibisah, (Navrongo Health Research Centre); Kadija Ouedraogo, Toussaint Rouamba, Karim Derra, Aminata Welgo, Halidou Tinto (Clinical Research Unit of Nanoro);Susan Goldstein, Winfreda Mdewa, Ijeoma Edoka (SAMRC Centre for Health Economics and Decision Science, PRICELESS); Mark Hanson, Caroline Fall, (Faculty of Medicine, University of Southampton); Emmanuel Cohen, Stephanie Wrottesley (SAMRC Developmental Pathways for Health Research Unit).

## Author Contributions

**Conceptualization:** Mary Barker, P. Romuald Boua, Adelaide Compaoré, Marion Danis, Maxwell Dalaba, Agnes Erzse, Karen J. Hofman, Engelbert A. Nonterah, Hermann Sorgho, Kate A. ward, Aviva Tugendhaft.

**Data curation:** Daniella Watson, P. Romuald Boua, Samuel Chatio, Adelaide Compaoré, Maxwell Dalaba, Agnes Erzse, Teurai Rwafa-Ponela, Aviva Tugendhaft.

**Formal analysis:** Daniella Watson, P. Romuald Boua, Adelaide Compaoré, Maxwell Dalaba, Agnes Erzse.

**Funding acquisition:** Kate A. ward.

**Investigation:** Daniella Watson.

**Methodology:** Daniella Watson, P. Romuald Boua, Adelaide Compaoré, Maxwell Dalaba, Agnes Erzse.

**Project administration:** Daniella Watson, Agnes Erzse, Polly Hardy-Johnson, Sarah H. Kehoe, Aviva Tugendhaft.

**Resources:** Kate A. ward.

**Software:** Daniella Watson.

**Supervision:** Daniella Watson, Mary Barker, Marion Danis, Maxwell Dalaba, Wendy T. Lawrence, Kate A. ward, Aviva Tugendhaft.

**Validation:** Daniella Watson.

**Writing – original draft:** Daniella Watson, Aviva Tugendhaft.

**Writing – review & editing:** Daniella Watson, Mary Barker, P. Romuald Boua, Samuel Chatio, Adelaide Compaoré, Marion Danis, Maxwell Dalaba, Agnes Erzse, Polly Hardy-Johnson, Sarah H. Kehoe, Karen J. Hofman, Wendy T. Lawrence, Engelbert A. Nonterah, Hermann Sorgho, Teurai Rwafa-Ponela, Kate A. ward, Aviva Tugendhaft.

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
