## [Decision Letter · Decision Letter 0]

1 Oct 2023

PONE-D-23-18772What works in engaging communities? Prioritising nutrition interventions in Burkina Faso, Ghana and South AfricaPLOS ONE

Dear Dr. Watson,

Thank you for submitting your manuscript to PLOS ONE. After careful consideration, we feel that it has merit but does not fully meet PLOS ONE’s publication criteria as it currently stands. Therefore, we invite you to submit a revised version of the manuscript that addresses the points raised during the review process.

We look forward to receiving your revised manuscript.

Kind regards,

Stephen Apanga, MD, MSc, PhD

Academic Editor

PLOS ONE

Journal Requirements:

 "This research was funded by the National Institute for Health Research (NIHR) (17\\63\\154) using UK aid from the UK Government to support global health research. The views expressed in this publication are those of the authors and not necessarily those of the NIHR or the UK Department of Health and Social Care. KJH and AE are supported by South African Medical Research Council/ Centre for Health Economics and Decision Science – PRICELESS SA, University of Witwatersrand School of Public Health, Faculty of Health Sciences, Johannesburg South Africa (D1305910-03)."  

Reviewers' comments:

Reviewer's Responses to Questions

**Comments to the Author**

1. Is the manuscript technically sound, and do the data support the conclusions?

Reviewer #1: Yes

Reviewer #2: Yes

2. Has the statistical analysis been performed appropriately and rigorously? 

Reviewer #1: N/A

Reviewer #2: Yes

3. Have the authors made all data underlying the findings in their manuscript fully available?

Reviewer #1: Yes

Reviewer #2: Yes

4. Is the manuscript presented in an intelligible fashion and written in standard English?

Reviewer #1: Yes

Reviewer #2: Yes

5. Review Comments to the Author

Reviewer #1: In this manuscript authors conducted a process evaluation of using the CHAT tool to encourage community engagement and participation in the prioritizing of nutrition interventions. The manuscript is well written and makes a contribution to what works in involving community members in prioritizing nutrition interventions to encourage a sense of ownership and sustainability of such interventions. However, I have a few issues I will like the authors to address.

• In the introduction section, line 120, authors have indicated that this is the first time the CHAT tool is used in the study setting. Although, this is agreeable, there is a published paper (Link here: https://pubmed.ncbi.nlm.nih.gov/36962493/) using CHAT tool in Navrongo, Ghana authored by some of the authors in the current manuscript. Although the work is cited in the manuscript, I will suggest it is cited earlier preferably in the background in which authors would explain the added value the current manuscript is bringing that is different from the already published work. What questions were unanswered in the published work from Navrongo, Ghana which are now being answered in the current manuscript? These should be well articulated in the background section.

• In line 220 of the methods section, check for spelling of HIV/AIDS

• In the methods section line 285, the site for selecting the participants in Soweto has the tendency of selection bias which may not be representative of the entire community. Authors could justify their choice of such an approach given that persons who do not seek for public transport from the hospital's taxi rank are obviously excluded. The approach is obviously different from the approach adopted in Navrongo and in Burkina Faso. Or they could cited it as a limitation of the study.

Reviewer #2: This is a well written paper which is a huge contribution to the limited literature in the field of deliberative pooling and especially in finding solutions to the challenges of maternal and child nutrition services in sub Saharan Africa.

Methods and materials

The word “identity” on line 130 should be “identify”

6. PLOS authors have the option to publish the peer review history of their article (what does this mean?). If published, this will include your full peer review and any attached files.

Reviewer #1: **Yes: **Victor Mogre

Reviewer #2: No

---

## [Author Response · Author response to Decision Letter 0]

23 Oct 2023

See response to reviewer document uploaded.

---

## [Editor Report · Decision Letter 1]

31 Oct 2023

What works in engaging communities? Prioritising nutrition interventions in Burkina Faso, Ghana and South Africa

PONE-D-23-18772R1

Dear Dr. Watson,

We’re pleased to inform you that your manuscript has been judged scientifically suitable for publication and will be formally accepted for publication once it meets all outstanding technical requirements.

Kind regards,

Stephen Apanga, MD, MSc, PhD

Academic Editor

PLOS ONE
---

## [Editor Report · Acceptance letter]

1 Dec 2023

PONE-D-23-18772R1 

What works in engaging communities? Prioritising nutrition interventions in Burkina Faso, Ghana and South Africa 

Dear Dr. Watson:

I'm pleased to inform you that your manuscript has been deemed suitable for publication in PLOS ONE. Congratulations! Your manuscript is now with our production department. 

Kind regards, 

on behalf of

Dr. Stephen Apanga 

Academic Editor

PLOS ONE